# Mineral Composition and Antioxidant Potential of Coffee Beverages Depending on the Brewing Method

**DOI:** 10.3390/foods9020121

**Published:** 2020-01-23

**Authors:** Katarzyna Janda, Karolina Jakubczyk, Irena Baranowska-Bosiacka, Patrycja Kapczuk, Joanna Kochman, Ewa Rębacz-Maron, Izabela Gutowska

**Affiliations:** 1Pomeranian Medical University in Szczecin, Department of Human Nutrition and Metabolomics, 24 Broniewskiego Street, 71-460 Szczecin, Polandkochmaan@gmail.com (J.K.); 2Department of Biochemistry, Pomeranian Medical University in Szczecin, 71 Powstańców Wlkp. Street, 70-111 Szczecin, Poland; irena.bosiacka@pum.edu.pl (I.B.-B.); patrycja2510@o2.pl (P.K.); 3Department of Vertebrate Zoology and Anthropology, Institute of Biology, University of Szczecin, 13 Wąska Street, 71-415 Szczecin, Poland; ewa.rebacz-maron@usz.edu.pl; 4Department of Medical Chemistry, Pomeranian Medical University in Szczecin, 71 Powstańców Wlkp. Street, 70-111 Szczecin, Poland; izagut@poczta.onet.pl

**Keywords:** beverages, brewing method, antioxidant potential, total polyphenols content, mineral composition

## Abstract

Coffee, being one of the world’s most popular beverages, is a rich source of dietary antioxidants. The aim of this study was to determine the mineral content and antioxidant activity as well as acidity of coffee beverages depending on the brewing technique. We tested coffee brews made and served at a popular urban coffee shop (Szczecin, Poland). Five coffee brewing techniques were used: Aeropress, drip, espresso machine, French press, and simple infusion. Our findings showed that the brewing method had a significant effect on all parameters tested in the study. The antioxidant activity of the beverages was high (31%–42% inhibition of DPPH (2,2-diphenyl-1-picrylhydrazyl); reduction potential from 3435.06 mol Fe3^+^/mL to 4298.19 mol Fe3^+^/mL). Polyphenolic content ranged from 133.90 g (French press) to 191.29 g of gallic acid/L (Aeropress brew), depending on the coffee extraction method. Mineral content was also found to differ between brewing methods. Coffees prepared by simple infusion and Aeropress provided a valuable source of magnesium, manganese, chromium, cobalt, and potassium, whereas the drip brew was found to be a good source of silicon.

## 1. Introduction

Along with changes in modern society, as people’s lives are getting faster and more intense, stimulant beverages are becoming increasingly popular. These products have been found to diminish fatigue, enhance concentration, boost productivity, and generally improve performance [1]. Coffee is one of such substances and, in parallel, is one of the most popular beverages [2]. For many people, coffee drinking is part of their lifestyle and a daily habit. Every day, millions of people—about 40% of the world population—begin their day with a morning cup of coffee [3]. Coffee beverages are consumed for various reasons, including their stimulatory effects resulting from the presence of caffeine, health benefits attributed to their rich phytochemistry, and, predominantly, due to excellent taste and aroma. Although flavor, aroma, and the content of caffeine (1,3,7-trimethylpurine-2,6-dione) play a role in its popularity, coffee (both beans and beverages) is a complex chemical mixture, reported to contain more than a thousand of different chemical compounds, including carbohydrates, lipids, nitrogen, vitamins, minerals, alkaloids, and phenolic compounds [2,3,4]. Various studies have suggested that coffee consumption helps to reduce the risk of number of health conditions, including Alzheimer’s disease, Parkinson’s disease, heart disease, type 2 diabetes mellitus, liver cirrhosis, as well as certain types of disorders [3,4]. Regular consumption of coffee, both green and roasted, is recommended in order to lower the risk of metabolic syndrome [5]. These health-promoting properties of coffee have been linked with its antioxidant content. Indeed, coffee is one of the richest sources of these compounds [4]. Coffee beverages are also a dietary sources of minerals [2,4,6]. The concentrations of minerals and biologically active substances depend of the place of origin of the beans, the degree/intensity of roasting, as well as the brewing method [5,6,7,8,9,10].

There are no reports in the literature linking the brewing method to the antioxidant and mineral content. In this study, an attempt was made to determine the correlation between the antioxidant activity and the content of minerals in coffee brews.

## 2. Materials and Methods

In this study, we tested coffee brews (espresso blend—100% arabica) made and served at a popular urban coffee shop (an outlet in Szczecin, Poland).

### 2.1. Coffee Brewing Methods

Filtered water was used to make coffee beverages.

Aeropress*:* The Aeropress coffee maker was used. For 250.0 mL of the infusion, 18.0 g of quite coarsely ground coffee was used. The water had a temperature of 93 °C, and the pressure was 2–4 bars. The brewing lasted 2 min.

Drip: A drip coffee maker device was used. Medium ground coffee beans (18.0 g) were placed in a paper filter. A total of 300.0 mL of water at 92 °C was added to the tank and the machine was turned on. After 2.5 min, when the coffee was ready, samples were taken for measurement.

Espresso: An espresso machine was used. For 250 mL of the brew, 17.0 g of the most finely ground coffee was used. The water had a temperature of 92 °C. The coffee machine was set to 9 bars with regards to pressure.

Simple infusion: A total of 17.0 g of very finely ground coffee was placed inside the beaker, 250 mL of hot water (92 °C) poured inside, and it was left to steep. After 5 min, coffee samples were taken for analysis.

French press: A French press device, also called a press pot or a coffee plunger, was used. The pot was placed on a flat surface, with the plunger pulled out, and 17.0 g of medium ground coffee was added and then 300.0 mL of hot water (92 °C) was gently poured inside. Then the plunger was reinserted into the pot on the surface of the beverage and plunged down after 5 min. Under these conditions, a pressure of 1–2 bars was reached. Once the press plunger was pushed down, coffee samples were taken for analysis.

### 2.2. Spectrophotometric Assay

Determination of antioxidant activity, the reduction potential, and polyphenol content were determined using spectrophotometer Agilent 8453UV. All assays were performed in triplicate. For analysis, infusions were diluted 20 times.

#### 2.2.1. Determination of Antioxidant Activity

The antioxidant activity of samples was measured with spectrophotometric method using synthetic radical DPPH (2,2-diphenyl-1-picrylhydrazyl, Sigma Aldrich, Darmstadt, Germany). Antioxidant potential (antioxidant activity, inhibition) of tested solutions has been expressed by the percent of DPPH inhibition [6].

#### 2.2.2. Determination of Total Polyphenol Content

Determination of polyphenols was performed according ISO (International Organization for Standarization) 14502-1 using Folin–Ciocalteu reagent [11]. A total of 5.0 mL of a 10% Folin–Ciocalteau solution and 1.0 mL of test sample were successively introduced into the vial. The sample was shaken vigorously, and after 5 min, 4.0 mL of 7.5% Na_2_CO_3_ solution was added. The prepared solution was incubated for 60 min at room temperature. Reference solution was prepared the same way, but distilled water was added instead of the tested sample. Absorbance at 765 nm was measured. Total polyphenols content (ppm) was determined from the calibration curve using gallic acid as the reference standard.

#### 2.2.3. Determination of the Reduction Potential by the FRAP (Ferric Reducing of Antioxidant Power) Method

The FRAP method, used to determine the total reduction potential, is based on the ability of the test sample to reduce Fe3^+^ ions to Fe2^+^ ions. The FRAP unit determines the ability to reduce 1 mole Fe3^+^ to Fe2^+^ [12]. Absorbance at 593 nm was measured.

### 2.3. Determination of Total Acidity by Titration

The total acidity of coffee beverages subjected to analysis was determined by titrating the sample with a standard solution of NaOH in the presence of phenolphthalein until the color changed to light pink and stayed pink for at least 30 s [13]. The result was reported in grams per 100 mL of the infusion and expressed in units of malic acid.

### 2.4. Determination of Mineral Content

Samples (coffee beans: *n* = 3; coffee beverages: *n* = 3 of each type of coffee drink) were analyzed using inductively coupled plasma optical emission spectrometry (ICP-OES, ICAP 7400 Duo, Thermo Scientific (Waltham, MA, USA) equipped with a concentric nebulizer and cyclonic spray chamber to determine their Ca, Fe, Mg, K, Na, Sr, Zn, and P content. Analysis was performed in radial and axial mode. The samples were mineralized using microwave digestion system MARS 5 (CEM, Matthews, NC, USA). The weight of solid samples given to analysis was at least 0.1 g. The volume of liquid samples was 0.75 mL. The samples were transferred to clean polypropylene tubes. Then, 4 mL of 65% HNO_3_ (Suprapur, Merck, Darmstadt, Germany) was added to each vial and each sample was allowed 30 min pre reaction time in the clean hood. After completion of the pre-reaction time, 1 mL of non-stabilized 30% H_2_O_2_ solution (Suprapur, Merck, Darmstadt, Germany) was added to each vial. Once the addition of all reagents was complete, the samples were placed in special Teflon vessels and heated in a microwaved digestion system for 35 min at 180 °C (15 min ramp to 180 °C and maintained at 180 °C for 20 min). At the end of digestion, all samples were removed from the microwave and allowed to cool to room temperature. In the clean hood, samples were transferred to acid-washed 15 mL polypropylene sample tubes. A further 20-fold (for solids) and 5-fold (for liquids) dilution was performed prior to ICP-OES measurement. The samples were spiked with an internal standard to provide a final concentration of 0.5 mg/L ytrium, 1 mL of 1% Triton (Triton X-100, Sigma), and diluted to the final volume of 10 mL with 0.075% nitric acid (Suprapur, Merck). Samples were stored in a monitored refrigerator at a nominal temperature of 4 °C until analysis. Blank samples were prepared by adding concentrated nitric acid to tubes without sample and subsequently diluted in the same manner described above. Multielement calibration standards (ICP multi-element standard solution IV, Merck, Germany; phosphorus standard for ICP, Inorganic Ventures, United States) were prepared with different concentrations of inorganic elements in the same manner as in blanks and samples. Samples of reference material (NIST SRM (National Institute of Standards and Technology-Standard Reference Material) 1486 bone meal; *n* = 3) were prepared in the same manner as the samples (Table 1). Deionized water (Direct Q UV, Merck Millipore Corporation, approximately 18.0 MΩ) was used for preparation of all solutions. The wavelengths (nm) were P 178.284, K 766.490, Ca 315.887, Fe 239.562, Zn 213.856, Sr 421.552, Na 589.592, and Mg 285.213.

### 2.5. Statistical Analysis

In all the experiments, three individually extracted samples were analyzed and all the analyses were carried out at least in triplicate (nine replications). The statistical analysis was performed using Stat Soft Statistica 13.0 (StatSoft Polska Sp. z o.o., Kraków, Poland) and Microsoft Excel 2017 (Microsoft, Poland). The results are expressed as mean values and standard deviation (SD). To assess the differences between examined parameters, one-way analysis of variance (ANOVA) and Tukey’s post-hoc test (StatSoft Polska Sp. z o.o., Kraków, Poland) were used. Differences were considered significant at *p* ≤ 0.05.

## 3. Results

The content of mineral elements in ground coffee beans is shown in Table 2.

However, from a practical point of view, infusions are products that can be considered in terms of health-promoting properties, or as sources of minerals in the daily diet.

The brews included in the study underwent quantitative and qualitative analysis for mineral content, which showed that the highest level of calcium (Ca) was found in coffee from the espresso machine (25.71 mg/L), and the lowest level was found in the drip brew (16.34 mg/L) (Figure 1a). The Ca concentrations in individual coffee preparations were significantly different, except the comparison between the French press and simple infusion (*p* = 0.7552).

Magnesium (Mg) content ranged from 77.15 mg/L (French press) to 116.30 mg/L (Aeropress) (Figure 1b). The differences in Mg content in the respective brews were statistically significant, except for drip vs. simple infusion (*p* = 0.0573).

The highest level of manganese (Mn) was found in Aeropress coffee (0.640 mg/L), and the lowest was found in the coffee from the French press (0.443 mg/L) (Figure 1c). The differences were mainly statistically significant. The exceptions were in the simple infusion vs. Aeropress (*p* = 0.4960), simple infusion vs. drip (*p* = 0.0878), and in the French press vs. espresso (*p* = 0.9532).

In the case of chromium (Cr), the highest content thereof was again found in Aeropress coffee (0.037 mg/L) (Figure 1d). The differences in the content of the mineral element between tested samples were statistically significant only regarding coffee from the espresso machine (vs. Aeropress, *p* = 0.0001; vs. drip, *p* = 0.0077; vs. simple infusion, *p =* 0.0039; vs. French press, *p* = 0.0232).

Coffee from the espresso machine was found to be the richest source of Zn (0.235 mg/L), the lowest level of which was determined in Aeropress coffee (0.123 mg/L) (Figure 2a). Statistically significant differences were observed for espresso vs. Aeropress (*p* = 0.0001), espresso vs. drip (*p* = 0.0002), espresso vs. simple infusion (*p =* 0.0002), and in espresso vs. French press (*p* = 0.0001).

In the analysis of copper (Cu) levels, it was demonstrated that coffee from the espresso machine was the best source of that mineral (0.085 mg/L), whereas in Aeropress and drip brews its levels were below the detection limit (Figure 2b). Statistically significant differences in Cu levels were found in Aeropress vs. espresso (*p* = 0.000129), Aeropress vs. simple infusion (*p* = 0.0072), drip vs. espresso (*p* = 0.0001), drip vs. simple infusion (*p* = 0.0072), espresso vs. simple infusion (*p* = 0.0017), and in French press vs. simple infusion (*p* = 0.0276).

In the case of iron (Fe), the lowest levels were observed in coffee from the French press (0.346 mg/L), and the highest in the drip brew (0.439 mg/L) (Figure 2c). Significant differences were found between the coffee from the French press and the remaining methods (vs. Aeropress, *p* = 0.0002; vs. drip, *p* = 0.0001; vs. espresso, *p* = 0.0001; vs. simple infusion, *p* = 0.0001).

The highest content of cobalt (Co) was determined in the coffee made by simple infusion (0.012 mg/L), and the lowest in coffee from the espresso machine (0.006 mg/L) (Figure 2d). The differences were statistically significant for the following coffee preparations: espresso vs. drip (*p* = 0.00039), simple infusion vs. Aeropress (*p* = 0.0494), simple infusion vs. espresso (*p* = 0.000136), French press vs. drip (*p* = 0.002342), and in French press vs. simple infusion (*p* = 0.000271).

The lowest levels of phosphorus (P) were observed in the coffee from the French press (49.64 mg/L), and the highest in the Aeropress coffee (81.58 mg/L) (Figure 3a). The differences between the respective coffee preparations were statistically significant, except for the difference between drip and simple infusion (*p* = 0.2176).

The lowest level of silicon (Si) was observed in the coffee from the French press (2.55 mg/L), and the highest (3.44 mg/L) in the drip coffee (Figure 3b). The majority of differences between the observed results were statistically significant. The exception was the insignificant difference in silicon content between French press and espresso (*p* = 0.8914).

Potassium (K) levels in coffee preparations ranged from 887.38 to 1540.70 mg/L (Figure 3c). The highest K content was found in Aeropress coffee, and the lowest was found in the coffee from the French press. All differences between these results were statistically significant (*p* = 0.0001; *p* = 0.0003).

In the case of sodium (Na), the highest levels were found in coffee made by simple infusion (27.810 mg/L), and the lowest were found in the drip brew (24.736 mg/L) (Figure 3d). Statistically significant differences in Na content were found for Aeropress vs. espresso (*p* = 0.00159), Aeropress vs. simple infusion (*p* = 0.0003), drip vs. espresso (*p* = 0.0002), drip vs. simple infusion (*p* = 0.0001), espresso vs. French press (*p* = 0.0421), and simple infusion vs. French press (*p* = 0.0076).

The coffee beverages made using different methods were also analyzed for the content of toxic elements (Figure 4). Starting with nickel (Ni), the drip brew was found to contain the highest levels thereof (0.0271 mg/L), and coffee from the French press had the lowest source of exposure (0.0213 mg/L). The following differences were statistically significant: simple infusion vs. drip (*p =* 0.0069), simple infusion vs. Aeropress (*p* = 0.0402), French press vs. drip (*p =* 0.0015), French press vs. espresso (*p =* 0.0397), and French press vs. Aeropress (*p =* 0.0098).

In the case of strontium (Sr), coffee from the Espresso machine was the predominant source of exposure (0.734 mg/L), and the lowest concentrations were found in the drip brew (0.278 mg/L). The differences in Sr content were statistically significant between brewing methods (*p =* 0.0001). 

The highest exposure to aluminum (Al) came from the drip coffee (1.55 mg/L), and the lowest concentrations were found in the brew from the espresso machine (0.31 mg/L). Statistically significant differences were found only for the drip brew vs. the remaining brewing methods (*p =* 0.0001).

The concentrations of molybdenum (Mo) and cadmium (Cd) in the coffee beverages were below the detection limit.

### 3.1. Analysis of Antioxidant Properties of Coffee Preparations

The antioxidant potential of the coffee preparations included in the study, expressed as DPPH inhibition percentage, ranged from 31% to 42% (Figure 5a). The lowest figure is representative of coffee from the espresso machine, and the highest refers to the Aeropress brew. The differences were statistically significant for the following samples: Aeropress vs. espresso (*p =* 0.0001), Aeropress vs. simple infusion (*p =* 0.0001), and Aeropress vs. French press (*p =* 0.0001), as well as drip vs. espresso, drip vs. simple infusion, and drip vs. French press (*p =* 0.0001).

The lowest total polyphenol content was found in the coffee from the French press (133.90 g gallic acid/L) and the highest was observed in the Aeropress brew (191.29 g gallic acid/L) (Figure 5b). The differences between the respective coffee brews were statistically significant, except for the difference between the coffee from the French press vs. espresso (*p =* 0.4733).

The reduction potential of the coffee brews ranged from 3435.06 mol Fe ^3+^/mL for the coffee from the French press to 4298.19 mol Fe ^3+^/mL in the Aeropress coffee (Figure 5c). The differences in the respective brews were statistically significant. The differences between Aeropress vs. drip (*p =* 0.5799) and espresso vs. French press (*p =* 0.2392) were demonstrated to be statistically insignificant.

The acidity of coffee brews ranged from 0.14% (espresso) to 0.6% (Aeropress) (Figure 5d). The only statistically insignificant differences in acidity were drip vs. French press (*p =* 0.4602), drip vs. simple infusion (*p =* 0.8514), and simple infusion vs. French press (*p =* 0.9617). The differences observed between the remaining brewing methods were statistically significant.

### 3.2. Correlations between the Analysed Parameters for Individual Brewing Methods

Statistical analysis of the results demonstrated significant positive correlations between the antioxidant potential parameters in the following brews:

-Between percentage DPPH inhibition and polyphenol content in Aeropress (*r* = 0.828), drip (*r* = 0.949), espresso (*r* = 0.752), and French press (*r* = 0.731). -Between DPPH and FRAP for Aeropress (*r* = 0.919), drip (*r* = 0.990), and espresso (*r* = 0.932).-Between polyphenol content and FRAP for Aeropress (*r* = 0.972), drip (*r* = 0.982), and espresso (*r* = 0.972). -The only correlation found in simple infusion was between DPPH and FRAP (*r* = 0.914).

A significant positive correlation between acidity and the levels of Mg, K, and P was observed in Aeropress, espresso, simple infusion, and French press brews (Table 3).

No significant correlations with acidity were found for the drip coffee. The individual coffee brews were also examined for statistically significant correlations between mineral levels. Individual brews were characterized by diverse, both positive and negative, correlations between mineral levels. The results are presented in Table 4.

## 4. Discussion

Coffee is one of the most widely consumed beverages in the world. It is known for its organoleptic qualities and is appreciated by countless coffee aficionados [2]. Because of its popularity, it attracts the interest of researchers, who continue to examine its impact on human health. The body of research has so far pointed to the safety of coffee consumption by the majority of social groups, as well as its positive impact on health. Studies have highlighted the relationship between coffee consumption and reduced risk for certain cancers, cardiovascular diseases, type 2 diabetes, and Parkinson’s disease [14]. However, the preparation method has a significant impact on many properties of the obtained brew [7]. For instance, it affects the aromatic compounds profile, acidity, fatty acid profile [15], caffeine and chlorogenic acid content [7,15], levels of diterpenic esters [16], furan [17], and isoflavones [18], as well as the extraction of biogenic and toxic elements into the coffee brew [10]. The biochemical properties of coffee preparations are determined by factors including brewing temperature, extraction time, and coffee grind size [7,16]. Recent research has indicated that the homogenization of the 30 ground coffee samples affected the extraction of lignans belonging to polyphenols. Comparison of lignan extraction yield in espresso coffee and ground coffee showed that these molecules seem to be completely extracted during espresso coffee percolation [19]. Judging by the present findings, it may be concluded that the levels of bioactive compounds and minerals in coffee also depend on the brewing method.

In present study, it was observed that the mineral composition changed depending on the brewing method, which may be of high importance from a nutritional point of view. The beverages included in the study were found to contain not only minerals that are essential for body functions, but also toxic elements.

Calcium plays a range of important functions in the body, and in order to maintain normal serum Ca levels, adults over the age of 25 require an intake of 750 mg/day for women and 950 mg/day for men [20]. Therefore, it seems that coffee may not be regarded as a significant source of Ca in the daily diet, as by consuming on average two cups of the beverage a day (approximately 360 mL) one can provide no more than about 1.3% and 1.0% (for women and men, respectively) of the daily requirement for that mineral (using the espresso machine). Magnesium acts as a cofactor for a vast number of enzymes, especially in energy metabolism. The daily requirement for Mg in an adult is 400 mg, and so consuming two cups of coffee provides about 7%–10% of the required amount [20]. Therefore, in nutritional terms, the coffee beverages included in the study may be regarded as a significant source of that mineral, predominantly prepared with simple infusion and Aeropress, with these being identified as the best brewing methods.

Taking into account the daily requirement for phosphorus, which in adults may be up to 550 mg [20], it was demonstrated that drinking Aeropress coffee provides approximately 3.2% of this mineral. 

Silicon is not essential for humans. The estimated typical dietary intake (20–50 mg silicon/day) corresponds to 0.3–0.8 mg/kg body weight per day in a 60 kg person. These intakes are unlikely to cause adverse effects [21]. In this context, drinking two cups of drip coffee accounts for between 2.4% and 6.1% of the daily silicon intake.

Other elements found in considerable quantities in the coffee beverages included in the study were manganese, chromium, cobalt, and potassium, with the highest quantities determined in the Aeropress coffee and in simple infusion coffee. The recommended dietary allowance (RDA) for manganese amounts up to 3.0 mg/day [20], and thus drinking two cups of Aeropress coffee may deliver approximately 7.7% of the daily requirement for this mineral, which functions as a cofactor for a variety of enzymes, including arginase, glutamine synthetase (GS), pyruvate carboxylase, and manganese superoxide dismutase (Mn-SOD). Through these metalloproteins, Mn plays critically important roles in development, digestion, reproduction, antioxidant defense, energy production, immune response, and regulation of neuronal activities [22]. In the case of Cr, RDA guidelines differ from country to country [23]. According to the World Health Organization (WHO), the average requirement for this mineral amounts to 0.035 mg/day [24], and thus two cups of Aeropress coffee would correspond to approximately 38% of this amount, making it a potentially significant source of chromium. 

The body of an adult person contains on average approximately 150 g potassium, and nearly 90% of that mineral is found inside cells. Potassium is responsible for maintaining normal water–electrolyte balance of the body and osmotic equilibrium in cells. Importantly, its functions include the activation of numerous enzymes and participation in carbohydrate and protein metabolism [25]. Serum levels of the mineral and overall body content make it impossible to determine its daily requirement, but it should be supplied in sufficient quantities to reduce the risk of diseases such as arterial hypertension, stroke, and ischaemic heart disease [19,23]. According to the experts from the European Food Safety Authority (EFSA) [20], the RDA for potassium should be 3500 mg/day, and thus coffee would provide approximately 15.8% of the daily K requirement for an adult. Considering the fact that potassium is present in practically all food products, with large quantities found in potatoes and cereal products, which are the most common food staples, the quantities supplied with coffee may account for a significant source of the mineral in the daily diet.

According to EFSA, evaluation of the average requirements for sodium and chloride is on-going [20]. The WHO, in turn, advises that the daily intake of sodium in adults should not exceed 2 g [24]. Consumption of two cups of the coffee brew that had the highest sodium content, that being the simple infusion, accounts for about 10.0% of that amount.

Cobalt and its compounds are widely distributed in nature and are part of numerous anthropogenic activities. Although cobalt plays a biologically necessary role as a metal constituent of vitamin B12 and is also needed to keep the body’s nervous system healthy, excessive exposure has been shown to induce various adverse health effects [26]. The average adult intake of cobalt is 5–8 µg/day, and even a single cup of coffee (180 mL) prepared by simple infusion may supply approximately 2.16 µg of the mineral, which accounts for up to 43% of the daily intake. Although a safe RDA for cobalt has not been established to date [26], the lethal dose of cobalt (LD50) is thought to be about 150–500 mg/kg of body weight [27].

Coffee beverages in our study were also analyzed for the content of zinc and copper, with low quantities observed for both minerals. Considering the RDA for Zn, which amounts to approximately 11 mg/day [20], it may be concluded that two cups of coffee from the espresso machine would deliver up to 0.8% of the daily requirement for that mineral and up to about 2% RDA for copper, for which the daily requirement is 1.5 mg/day [20]. Iron levels in the analyzed beverages fell in a similar range. Taking into account the daily requirement for Fe, amounting to 11 mg [20], the intake of the mineral in the drip brew corresponded to approximately 1.4% RDA.

Some of the minerals supplied to the body with the diet are required in large amounts to ensure normal body function (e.g., Ca and Mg), whereas others, such as Cu, Zn, K, Na, Fe, Cr, and Co, are only needed in small concentrations. However, there are also some elements, for example, strontium or aluminum, whose functions in the body are unknown [28]. In the case of toxic elements, their content in the diet should preferably stay below certain levels, due to the risk of adverse effects in response to excessive exposure. The presence of aluminum in soil is a natural phenomenon, but current studies suggest that the natural environment is significantly exposed to elevated levels of this mineral [29], whereas excessive accumulation of aluminum in brain tissue manifests itself primarily by memory deficit and posture dysfunction [29]. Vegetables cultivated in areas contaminated with heavy metals may include large amounts of aluminum, but it seems that the main source of human exposure comes from the public water supply [29,30]. For the general population, the intake of aluminum from food is 7.2 mg/day for females and 8.6 mg/day for males on average [31]. According to our findings, drinking two cups of coffee with the highest Al content, this being the drip brew (1.55 mg/L), supplies 0.55 mg of the mineral, and so the beverage does not constitute a significant source of aluminum in daily diet.

Nutritional requirements or RDA for nickel have not been established. The role and adverse health effects of nickel and nickel compounds have been assessed by several authors [32,33,34], but the role of nickel in the physiology of living organisms has not been fully explained. Most likely, Ni in animals and humans participates in erythropoiesis by influencing vitamin B12 metabolism and is regarded as a trace element in nutrition. Although there have been no reports in the literature to date on nickel deficiency, it is good to remember that nickel-rich foodstuffs include lentils, cocoa, chocolate, and nuts [35]. The daily intake of nickel from the average diet is about 150 μg, which corresponds to 2.5 μg/kg body weight per day in a 60 kg adult. Consumption of foods with a high nickel content and additional exposure from first-run drinking water and kitchen utensils may result in intake higher than the critical dose of this substance [21]. In terms of epidemiology, the most critical nickel-related issue to human health is nickel allergy. In those with a strong response, dermatitis may be caused by oral administration of 300 µg of nickel. This amount corresponds to doubled daily intake of this element [35]. In our study, Ni levels determined in coffee beverages made using different methods were more or less on a par. Two cups of coffee could supply about 6.5% of the daily intake of this element.

With respect to strontium, coffee from the espresso machine was the highest source of exposure (0.734 mg/L), and the lowest concentrations were found in the drip brew (0.278 mg/L). The Environmental Protection Agency (EPA) recommends that drinking water levels of stable strontium should not be more than 4 mg/L [36]. In the light of the aforementioned recommendation, coffee beverages contain considerable amounts of strontium, and the consumption of two cups a day may supply even 6.6% of this element. 

The data obtained in this study are mostly consistent with other studies regarding elemental composition of coffees [10,37]. Potassium is the most abundant element in coffee beans and also in brews mainly due to its high water solubility [37,38]. Other studies show that the concentration of Ca in coffee beans is indeed high (about 1400 mg/kg); however, the percentage of this element travelling from the beans to the infusion is low, similarly to P and Mg. Debastiani et al. [37] reported that levels of Ca, K, P, and Mg in the drinking coffee were similar to our results, and the authors suggested that coffee is a good source of these substances [37]. Low levels of Si, Ca, Fe, Cu, and Zn in brews, despite their presence in coffee beans and their higher concentrations in spent coffee, suggest that coffee behaves as a sponge by absorbing part of these elements from the hot water [10,37]. The concentrations of Mn, Cu, Zn, and Fe in the present research were similar to other studies; however, we found lower levels of Al (except in drip coffee) in our coffee samples [37].

The very high polyphenolic content found in the studied coffee beverages suggests that they are a rich source of biologically active compounds. Polyphenolics are credited with a broad range of health-promoting properties, including antioxidant, antimicrobial, anti-inflammatory, prebiotic, and anticarcinogenic activity [39]. Phenolic acids have been also demonstrated to be inversely associated with hypertension [40]. The most abundant and important polyphenols found in coffee are chlorogenic acids [41]. According to research findings, these compounds have anti-diabetic, anti-inflammatory, anti-carcinogenic, and anti-obesity effects [42]. In our study, the lowest total polyphenol content was found in the coffee from the French press (133.90 g gallic acid/L) and the highest was observed in the Aeropress brew (191.29 g gallic acid/L). These results are similar or lower than those reported by other authors but the infusions were made by traditional methods [43,44,45]. In soft and hard infusion, the total phenolic content for Arabica coffee was in the range of 94 to 96 mg QE/g of coffee, and in French Press was 100.78 mg QE/g of coffee [45]. Out of all coffee brewing methods investigated in the present study, the highest antioxidant potential, polyphenol content, and redox potential was observed in the brew made in the Aeropress. It was also characterized by the highest acidity. These findings lend themselves to the conclusion that this brewing method is responsible for the highest content of health-promoting compounds in a coffee beverage.

The antioxidant potential of coffee preparations is also affected by several factors, including the brewing method and bean variety. Wolska et al. [6] investigated percentage DPPH inhibition in Robusta, Arabica, and green coffee beans subjected to brewing using five different methods. The antioxidant activity of the beverages was high—71.97% to 83.21% inhibition of DPPH depending on bean type and extraction method. Ramadan-Hassanien [46] studied the antioxidant potential of instant coffee, Turkish coffee, and cappuccino. Cappuccino had the highest antioxidant activity—66.0% inhibition of DPPH—with instant coffee producing merely 14.0% inhibition of DPPH. A study by Caporaso et al. [47], which included a comparison of the antioxidant potential of coffee preparations made using different methods (espresso, moka, and American brews) demonstrated that the brewing technique has a significant effect on the chemical composition and quality of the resulting beverage. Please note that the literature is elusive regarding Aeropress brewing.

This study confirmed that the type of coffee preparation has a significant impact on the antioxidant potential of coffee beverages. According to Vicente et al. [48], coffee beverages can reduce the consequences of oxidative stress in the human body. Our results confirm that coffee infusions are a rich source of antioxidant compounds.

In conclusion, it was demonstrated that coffee brewing technique has a significant effect on both the antioxidant potential of the beverage and the levels of specific minerals. Coffee prepared by simple infusion and Aeropress provided a valuable source of magnesium, manganese, chromium, cobalt, and potassium, whereas the drip brew was found to be a good source of silicon. In addition, coffee was found to be a rich source of polyphenols with powerful antioxidant properties.

## Figures and Tables

**Figure 1 foods-09-00121-f001:**
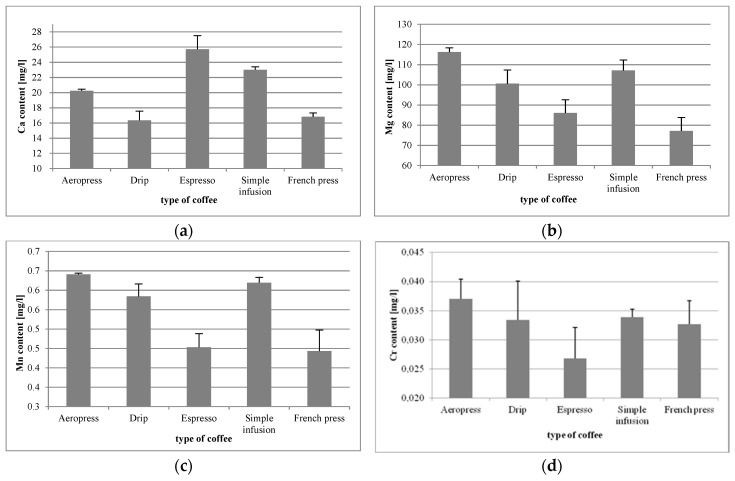
Concentrations of Ca (**a**), Mg (**b**), Mn (**c**), and Cr (**d**) in coffee beverages made using different methods.

**Figure 2 foods-09-00121-f002:**
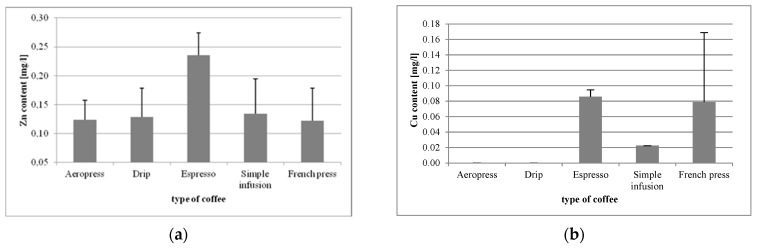
Concentrations of Zn (**a**), Cu (**b**), Fe (**c**), and Co (**d**) in coffee beverages made using different methods.

**Figure 3 foods-09-00121-f003:**
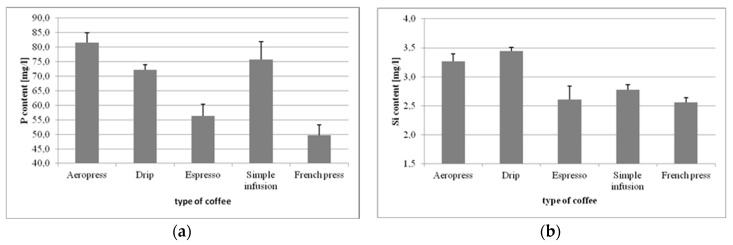
Concentrations of P (**a**), Si (**b**), K (**c**), and Na (**d**) in coffee beverages made using different methods.

**Figure 4 foods-09-00121-f004:**
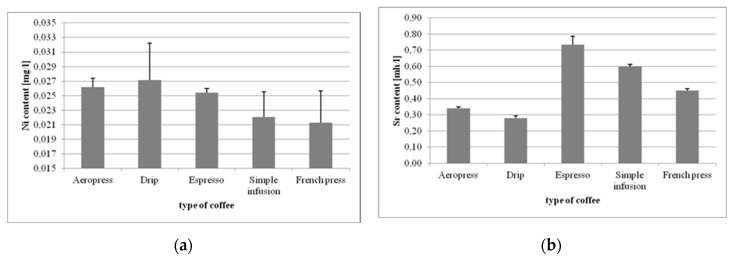
Concentrations of Ni (**a**), Sr (**b**), and Al (**c**) in coffee beverages made using different methods.

**Figure 5 foods-09-00121-f005:**
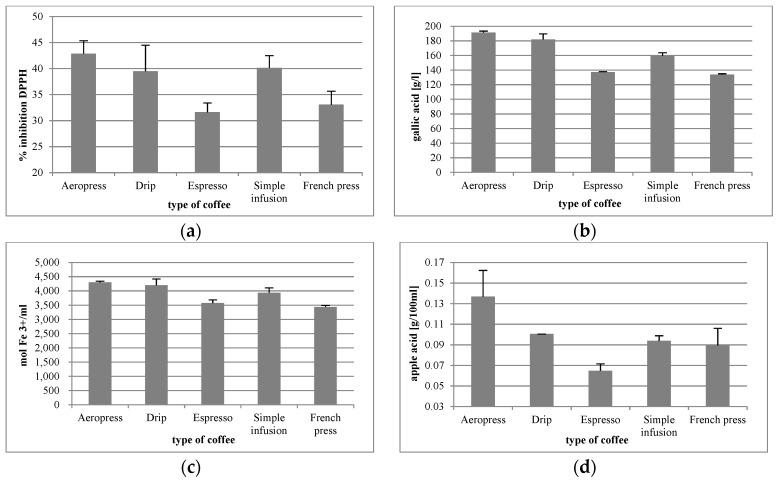
Antioxidant potential (**a**), total polyphenol content (**b**), reduction potential (**c**), and acidity (**d**) of coffee beverages made using different methods. DPPH: 2,2-diphenyl-1-picrylhydrazyl.

**Table 1 foods-09-00121-t001:** Analysis of reference material bone meal NIST-SRM (National Institute of Standards and Technology-Standard Reference Material)1486 using inductively coupled plasma optical emission spectrometry (ICP-OES).

Chemical Element	NIST-SRM 1486 Certified (mg/kg)	NIST-SRM 1486 Measured (mg/kg)	Recovery (%)
P 178.284	123,000.00	131,500.00	107%
K 766.490	412.00	417.00	101%
Ca 315.887	265,800.00	247,021.00	93%
Fe 239.562	99.00	102.15	103%
Zn 213.856	147.00	143.60	98%
Sr 421.552	264.00	258.17	98%
Na 589.592	5000	4826.33	97%
Mg 285.213	4600	4382.13	95%

**Table 2 foods-09-00121-t002:** Mineral composition of coffee beans.

Elements	Mineral Content (mg/kg)	
	Mean	SD
**Ca**	1441.20	49.49
**Fe**	48.36	3.30
**Mg**	2133.91	43.42
**K**	18,634.66	538.67
**Na**	83.75	10.01
**Sr**	16.25	0.80
**Zn**	9.93	0.42
**P**	2154.23	41.80

**Table 3 foods-09-00121-t003:** Statistically significant (at *p* ≤ 0.05) correlation (*r*) between elements in coffee: (**a**) Aeropress, (**b**) drip, (**c**) espresso, (**d**) simple infusion, and (**e**) French press.

**(a) Aeropress**		
**Correlations (*r*) between Elements**		
	**Positive**	**Negative**
Ca and	Zn; Co	Fe; Cr; Ni
Mg and	K; Na; Co; Sr; Si	Fe; Cr; Ni; Al
Mn and	Zn; Al	P; Na; Sr; Si
P and	K; Na; Sr; Si	Zn; Al
K and	Na; Co; Sr; Si	Fe; Cr; Ni; Al
Zn and		Fe
Fe and	Cr; Ni	Co
Na and	Co; Sr; Si	Cr; Al
Cr and	Ni; Al	Co; Sr; Si
Ni and		Co
Co and	Sr	
Sr and	Si	Al
Al and		Si
**(b) Drip**		
Ca and	Mg; Mn; P; K; Na; Co; Al; Si	
Mg and	Mn; P; K; Na; Co; Sr; Al; Si	
Mn and	P; K; Na; Co; Al; Si	Zn; Ni
P and	K; Na; Co; Sr; Al; Si	
K and	Na; Co; Sr; Al; Si	
Zn and	Fe; Cr; Ni	
Fe and	Cr; Ni; Sr	
Na and	Co; Al; Si	
Cr and	Ni; Sr	
Co and	Sr; Al; Si	
Sr and	Al; Si	
Al and	Si	
**(c) Espresso**		
Ca and	Mg; Mn; P; K; Na; Ni; Sr; Si	Zn; Cr; Co; Al
Mg and	Mn; P; K; Na; Ni; Sr; Si	Zn; Fe; Co; Al; Cu
Mn and	P; K; Na; Ni; S; Si	Zn; Cr; Co; Al
P and	K; Na; Ni; Sr Si	Zn; Fe Co; Al Cu
K and	Na; Ni; Sr; Si	Zn; Fe; Cr; Co; Al
Zn and	Cr; Al	Na; Ni; Sr; Si
Fe and	Co; Cu	Na; Sr
Na and	Ni; Sr; Si	Cr; Co; Al
Cr and	Al (*r* = 0.875)	Ni; Si
Ni and	Sr; Si	Al
Co and	Al; Cu	Sr; Si
Sr and	Si	Al
Al and		Si
**(d) Simple infusion**		
Ca and	Zn; Co	Fe; Cr; Ni
Mg and	K; Na; Co; Sr; Si	Fe; Cr; Ni; Al
Mn and	Zn; Al	P; Na; Sr; Si
P and	K; Na; Sr; Si	Zn; Al
K and	Na; Co; Sr; Si	Fe; Cr; Ni; Al
Zn and		Fe
Fe and	Cr; Ni	Co
Na and	Co; Sr; Si	Cr; Al
Cr and	Ni; Al	Co; Sr; Si
Ni and		Co
Co and	Sr	
Sr and	Si	Al
Al and		Si
**(e) French press**		
Ca and	Cr; N; Sr	
Mg and	P; K; Na; Ni; Si; Cu	
Mn and	P; K; Na	Zn; Fe; Cr; Sr
P and	K; Na; Ni; Si; Cu	Zn; Fe; Al
K and	Na; Ni; Si; Cu	Al
Zn and	Fe; Cr; Sr; Al	Na
Fe and	Cr; Al	
Na and	Ni; Si; Cu	
Cr and	Co; Sr; Al	
Ni and	Co; Si; Cu	
Co and	Sr	
Sr and	Al	
Si and	Cu	

**Table 4 foods-09-00121-t004:** Statistically significant (at *p* ≤ 0.05) correlation (*r*) between mineral content and acidity in coffee infusions.

Correlations (*r*) between Mineral Content and Acidity:		
	Positive	Negative
(a) in Aeropress coffee infusion		
acidity and	Mg; P; K; Na; Sr; S	Mn; Al
(b) in espresso coffee infusion		
acidity and	Mg; P; K; Na	Fe; Co; Cu
(c) in simple infusion coffee		
acidity and	Ca; Mg; P; K; Fe; Ni	Co
(d) in French press coffee infusion		
acidity and	Mg; Mn; P; K; Na	Zn; Fe; Cr; Sr; Al

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
