# Peer review of "Mineral Composition and Antioxidant Potential of Coffee Beverages Depending on the Brewing Method"

_foods, 2020, doi:10.3390/foods9020121_

Round 1
Reviewer 1 Report
The paper by Janda et al. reported the mineral composition and the level of polyphenols and antioxidant activity in Arabica coffee depending on the brewing methods.
Some comments:
Paragraph 2.2.1. It would be more correct to express the antioxidant activity data as equivalent of a standard solution (i.e. Trolox equivalent) rather than in % Paragraph 2.2.2. Why the authors choose gallic acid as standard? Coffee is rich of chlorogenic acids and the use of caffeoylquinic acid as reference compounds is more appropriate. The title of paragraph 2.4 is wrong. Please correct it. Paragraph 2.5. The authors stated that for each analysis three samples were analyzed. It is not clear if they carried out analysis on three individually extracted samples or on three aliquots of the same extracted sample. Please clarify this point. The discussion needed to be updated by comparing the authors data (such as mineral and polyphenolic content in coffee) with that previously reported in literature
Author Response
Dear reviewer,
I am very sorry for the mistake. Answers to reviewers were exchanged when sending files.
I attach the correct file. I am very, very sorry.
Thank you very much for all comments from reviewers. The changes have been applied to the manuscript. A language check was also made by a native speaker from Canada.
1)
The paper by Janda et al. reported the mineral composition and the level of polyphenols and antioxidant activity in Arabica coffee depending on the brewing methods.
Some comments:
Paragraph 2.2.1. It would be more correct to express the antioxidant activity data as equivalent of a standard solution (i.e. Trolox equivalent) rather than in %
Thank you, the information is very reasonable. Many studies use different units when it comes to the DPPH method (% inhibition, conversion to trolox or IC 50). We are in the process of introducing the trolox method as it helps to compare other test results and creates less restrictions.
Paragraph 2.2.2. Why the authors choose gallic acid as standard?
Chlorogenic acid is the acid most commonly found in coffee, however, the study did not perform a qualitative analysis of polyphenols but only quantitative. Gallic acid is used as often as gallic acid in coffee research. However, in the further plans, such an analysis is also planned. To compare polyphenols with other raw materials we are working on, the most commonly used gallic acid was used. However, we will try to include it in the next study.
Coffee is rich of chlorogenic acids and the use of caffeoylquinic acid as reference compounds is more appropriate. changed
The title of paragraph 2.4 is wrong. Please correct it. Changed - Determination of mineral content
Paragraph 2.5. The authors stated that for each analysis three samples were analyzed. It is not clear if they carried out analysis on three individually extracted samples or on three aliquots of the same extracted sample. Please clarify this point.
In all the experiments, three individually extracted samples were analysed and all the analysis were carried out at least in triplicate (9 replications).
The discussion needed to be updated by comparing the authors data (such as mineral and polyphenolic content in coffee) with that previously reported in literaturę
some new work has been added [37, 38, 40, 43, 44, 45]
Reviewer 2 Report
I have concern regarding reporting on the polyphenol content of the coffee samples. In the result section authors report solely on the gallic acid content and don’t report for example on chlorogenic acid content (which is the most common polyphenol find in coffee). I think since you used gallic acid as a reference compound and you did not aim to determine individual polyphenol content you should specify it, for example “determination of total polyphenol content” instead of “determination of polyphenols content”.
Line 397-399 phenolic acids have been also demonstrated to be inversely associated with hypertension (PMID: 28953227)
Authors should discuss their results in the context of the recently published paper quantifying lingers content in 30 different coffee samples (PMID: 31170854).
Author Response
Thank you very much for all comments from reviewers. The changes have been applied to the manuscript. A language check was also made by a native speaker from Canada.
2)
The paper by Janda et al. reported the mineral composition and the level of polyphenols and antioxidant activity in Arabica coffee depending on the brewing methods.
Some comments:
Paragraph 2.2.1. It would be more correct to express the antioxidant activity data as equivalent of a standard solution (i.e. Trolox equivalent) rather than in %
Thank you, the information is very reasonable. Many studies use different units when it comes to the DPPH method (% inhibition, conversion to trolox or IC 50). We are in the process of introducing the trolox method as it helps to compare other test results and creates less restrictions.
Paragraph 2.2.2. Why the authors choose gallic acid as standard?
Chlorogenic acid is the acid most commonly found in coffee, however, the study did not perform a qualitative analysis of polyphenols but only quantitative. Gallic acid is used as often as gallic acid in coffee research. However, in the further plans, such an analysis is also planned. To compare polyphenols with other raw materials we are working on, the most commonly used gallic acid was used. However, we will try to include it in the next study.
Coffee is rich of chlorogenic acids and the use of caffeoylquinic acid as reference compounds is more appropriate. changed
The title of paragraph 2.4 is wrong. Please correct it. Changed - Determination of mineral content
Paragraph 2.5. The authors stated that for each analysis three samples were analyzed. It is not clear if they carried out analysis on three individually extracted samples or on three aliquots of the same extracted sample. Please clarify this point.
In all the experiments, three individually extracted samples were analysed and all the analysis were carried out at least in triplicate (9 replications).
The discussion needed to be updated by comparing the authors data (such as mineral and polyphenolic content in coffee) with that previously reported in literaturę
some new work has been added [37, 38, 40, 43, 44, 45]
Round 2
Reviewer 2 Report
My comments has been completely omitted, moreover I did not receive any authors responses, thus I send you back the comments.
Comments:
I have concern regarding reporting on the polyphenol content of the coffee samples. In the result section authors report solely on the gallic acid content and don’t report for example on chlorogenic acid content (which is the most common polyphenol find in coffee). I think since you used gallic acid as a reference compound and you did not aim to determine individual polyphenol content you should specify it, for example “determination of total polyphenol content” instead of “determination of polyphenols content”.
Line 397-399 phenolic acids have been also demonstrated to be inversely associated with hypertension (PMID: 28953227)
Authors should discuss their results in the context of the recently published paper quantifying lingers content in 30 different coffee samples (PMID: 31170854).
Author Response
Dear reviewer,
I am very sorry for the mistake. Answers to reviewers were exchanged when sending files. I attach the correct file. I am very, very sorry.
Thank you very much for all comments from reviewers. The changes have been applied to the manuscript. A language check was also made by a native speaker from Canada.
2)
I have concern regarding reporting on the polyphenol content of the coffee samples. In the result section authors report solely on the gallic acid content and don’t report for example on chlorogenic acid content (which is the most common polyphenol find in coffee). I think since you used gallic acid as a reference compound and you did not aim to determine individual polyphenol content you should specify it, for example “determination of total polyphenol content” instead of “determination of polyphenols content”.
Chlorogenic acid is the acid most commonly found in coffee, however, the study did not perform a qualitative analysis of polyphenols but only quantitative. Gallic acid is used as often as gallic acid in coffee research. However, in the further plans, such an analysis is also planned. To compare polyphenols with other raw materials we are working on, the most commonly used gallic acid was used. However, we will try to include it in the next study.
We also changed polyphenols content to total polyphenol content in manuscript.
Line 397-399 phenolic acids have been also demonstrated to be inversely associated with hypertension (PMID: 28953227) added
Authors should discuss their results in the context of the recently published paper quantifying lingers content in 30 different coffee samples (PMID: 31170854). added
Recent research indicates that the homogenization of the 30 ground coffee samples affects the extraction of lignans belonging to polyphenols. Comparison of lignan extraction yield in espresso coffee and ground coffee showed that these molecules seem to be completely extracted during espresso coffee percolation.